

# An immunoinformatics approach for the design of a multi-epitope subunit vaccine for urogenital schistosomiasis

Olugbenga S. Onile[1], Adeyinka I. Fadahunsi[1], Ameerah A. Adekunle[1], Bolaji F. Oyeyemi[2] and Chiaka I. Anumudu[3]

[1] Biotechnology Programme, Department of Biological Sciences, Elizade University, Ilara-Mokin, Ondo State, Nigeria
[2] Molecular Biology Group, Department Science Technology, The Federal Polytechnic, Ado-Ekiti, Ado-Ekiti, Ekiti State, Nigeria
[3] Cellular Parasitology Programme, Department of Zoology, University of Ibadan, Ibadan, Oyo State, Nigeria

## ABSTRACT

Discovery of T and B memory cells capable of eliciting long-term immunity against schistosomiasisis is important for people in endemic areas. Changes in schistosomes environment due to developmental cycle, induces up-regulation of Heat Shock Proteins (HSPs) which assist the parasite in coping with the hostile conditions associated with its life cycle. This study therefore focused on exploring the role of HSPs in urogenital schistosomiasis to develop new multi-epitope subunit vaccine against the disease using immunoinformatic approaches. The designed subunit vaccine was subjected to in silico antigenicity, immunogenicity, allergenicity and physicochemical parameters analysis. A 3D structure of the vaccine construct was predicted, followed by disulphide engineering for stability, codon adaptation and in silico cloning for proper expression and molecular protein–protein docking of vaccine construct in the vector against toll-like receptor 4 receptor, respectively. Consequently, a 493 amino acid multi-epitope vaccine construct of antigenicity probability of 0.91 was designed. This was predicted to be stable, non-allergenic in nature and safe for human use.

Corresponding author
Olugbenga S. Onile,
olugbenga.onile@elizaeuniversity.edu.ng

## INTRODUCTION

Urinary schistosomiasis has been identified as the main cause of urogenital disease, and the main cause of bladder cancer in developing countries (*Onile et al., 2016*; *Rollinson et al., 2013*). In 76 countries, more than 200 million people are estimated to be affected by urogenital schistosomiasis with more than 100 million being urogenital schistosomiasis. In Sub-Saharan Africa, mortality rate of the disease is estimated at 250,000 persons per year (*Masamba et al., 2016*). In the year 2008, estimated cases of 29 million infected persons were reported in Nigeria which amount to the largest number of infections in Africa (*Adenowo et al., 2015*) and may have increased (*Abdulkadir et al., 2017*). *Bergquist (2013)* reported the spread of the disease to regions that were previously non-endemic.

These include reported cases of urinary schistosomiasis in adolescent refugee from Gambia (*Poddighe et al., 2016*) and 214 male migrant patients *Milesi et al. (2019)* all in Italy. *Uysal et al. (2014)* also reported a rare case of a 22-year-old Nigerian with urinary schistosomiasis in Turkey. Causes of several form of bladder pathologies and tumours in Africa including Nigeria has been linked to urogenital schistosomiasis (*Adebayo et al., 2017*; *Barsoum, 2013*; *Onile et al., 2016*). Praziquantel has been the most used drug for treating schistosomiasis since the early 2000s (*Chevalier et al., 2016*; *Doenhoff, Cioli & Utzinger, 2008*), but the drug has been found to be less effective against the worm in all stages of its life cycle and would not eliminate immature worms within 3–4 weeks of infection (*Alsaqabi & Lotfy, 2014*; *Aragon et al., 2009*; *Cioli et al., 2014*; *Doenhoff, Cioli & Utzinger, 2008*). Also, several cases of drug resistance to praziquantel due to pressure from continuous mass drug administration have now been reported (*Couto et al., 2011*; *Fallon et al., 1996*; *Melman et al., 2009*).

By transiting between intra-mammalian, aquatic and snail stages to develop into full maturity, schistosomes face a number of hostile environments throughout their lifecycle (*Mbah et al., 2013*) and heat shock responses have been associated with cellular stress during the movement of parasite from a cooler, low saline and freshwater environment to a warmer, saline environment of a human host (*Devaney, 2006*; *Mbah et al., 2013*).

Heat shock proteins (HSP) are an important group of molecules involved in a range of biological and developmental processes in schistosomes and other parasites (*Braschi et al., 2006*). Hsp 90 and other HSPs are chaperone proteins that assist proper folding of other proteins. They stabilize proteins against heat stress, and alos aid protein degradation. Other functions include stabilizing plethora of proteins required for tumour growth; therefore HSP inhibitors are being investigated as vaccines and anti-cancer drugs (*Das et al., 2019*). Identification of HSPs in cercarial gland secretions and the highest abundance of the protein transcripts in newly transformed schistosomula have been reported (*Ishida & Jolly, 2016*). The role of heat shock response in snail susceptibility to Schistosoma infection in the intermediate molluscan host has also been emphasized (*Ishida & Jolly, 2016*). *Onile et al. (2017)*, had reported several HSPs as possible biomarkers for the diagnosis of urinary schistosomiasis, they are also known to play critical roles in innate immunity and adaptive immunity. HSPs can activate specific toll-like receptors (TLRs), provide polypeptides for specific triggering of the acquired immune response and, play a major role in cross-presentation of extracellular antigens. This would result in the induction of CD8+ cytotoxic T-lymphocyte responses (*McNulty et al., 2013*). Antigen presenting cells (APCs) are known for recognizing a wide range of molecular patterns expressed by pathogens called pathogen-associated molecular patterns (PAMPs) during innate immune responses (*El-Din, 2016*; *Venugopal, Nutman & Semnani, 2009*). TLRs and NOD-like receptors through signaling pathways that induce upregulation of IFNs and other inflammatory cytokines are used by APCs to recognize these PAMPs (*Venugopal, Nutman & Semnani, 2009*). *Zhang et al. (2011)* had reported mice with deficient TLR after infection with *S. japomicum* showed increasing level of parasite egg load.

Heat shock proteins were chosen in this study as target for the development of a new multi-epitope subunit vaccine against urinary schistosomiasis for their role in Schistosoma biology.

## METHODOLOGY

### Retrieval of *Schistosoma haematobium* HSP sequence for vaccine construction

The amino acid sequences of *S. haematobium* HSPs were retrieved from the NCBI protein database (www.ncbi.nlm.nih.gov) and subjected to multiepitope vaccine designing. To test for the proteins ability to induce immune response within a host body, all the retrieved protein sequences were subjected to an antigenicity test using the ANIGENpro database (scratch.proteomics.ics.uci.edu). Antigenic probability of ≥0.8 was used to determine which proteins to be chosen for the next step of the multi epitope vaccine construct (*Pandey, Bhatt & Prajapati, 2018*).

### Prediction of cytotoxic T-lymphocytes and helper T-lymphocytes epitope and immunogenicity

To get an immunogenic cytotoxic T-lymphocytes (CTL) epitope with potential of inducing cell mediated immunity and form memory cells, all the three highly antigenic HSP sequences in FASTA format were fed into the NetCTL 1.2 server (http://www.cbs.dtu.dk/services/NetCTL/) to predict the CTL epitope at threshold score of 0.75 (using the default setting). The server predicts CTL epitopes from inputted protein sequences based on the training dataset (prediction of MHC-I binding peptides, proteasomal C-terminal cleavage, and Transporter Associated with Antigen Processing (transport efficiency) and only epitopes with combined score greater than 0.75 were selected as CTL epitopes (*Khatoon, Pandey & Prajapati, 2017*; *Pandey, Bhatt & Prajapati, 2018*). The selected CTL epitopes (Scores >0.75) were further analysed in Immune Epitope Design Database (www.iedb.org) for MHC class I immunogenicity prediction. The immunogenicity score determines the probability of eliciting an immune response (higher score means greater probability of immune response and vice versa) (*Pandey, Bhatt & Prajapati, 2018*).

The helper T-Lymphocytes (HTL) epitopes of 15mer length for mouse allele (H2-IAb, H2-IAd and H2-IEd) were predicted for the HSPs of *S. haematobium* using the Immune Epitope Design Database. The output epitopes were ranked based on lower percentile rank scores that is, the lower the ranked score the higher the binding affinity for HTL receptors (*Khatoon, Pandey & Prajapati, 2017*). Only epitopes with higher binding affinity (≤1.5) with MHC-II were selected for the final multi-epitope vaccine construct. In order to determine the ability of the predicted epitopes in activating Th1 type immune response followed by IFN-γ production, the top 17 HTL epitopes were subjected in FASTA format to the IFN epitope server by using the Motif and SVM hybrid as approach selection and IFN-γ versus other cytokine as model of prediction.
## Construction of multi-epitope subunit vaccine

To design a viable vaccine construct capable of inducing innate and adaptive immune response, the high scoring CTL and HTL epitopes from the present immunoinformatics predictions were used. These epitopes were joined together using AAY and GPGPG linkers which were added at the intra-epitope position thereby linking the CTL and HTL epitopes respectively (*Khatoon, Pandey & Prajapati, 2017*). Also, a TLR-4 agonist (RS-09; Sequence: APPHALS) was used adjuvant (*Pandey, Bhatt & Prajapati, 2018*; *Shanmugam et al., 2012*) and linked with the multi-epitopes (CTL and HTL) using EAAAK linker (*Lee et al., 2014*).

## B cell epitopes prediction for *S. haematobium* proteins

The BCPREDS server (ailab.ist.psu.edu/bcpred) was used to predict the linear B-cell epitopes for the final vaccine construct. The 493 amino acid sequence of the final vaccine construct was inputted into the BCPREDS server in plain format followed by the selection of a fixed length epitope prediction method and length of the epitope. The default method of BCPREDS was selected as the prediction method for the amino acid epitopes (*El-Manzalawy, Dobbs & Honavar, 2008*; *Pandey, Bhatt & Prajapati, 2018*). The specificity threshold was set as default at 75% and conformational epitopes were predicted using the ElliPro server for the tertiary protein structure of the vaccine construct (*Pandey, Bhatt & Prajapati, 2018*).

## Prediction of antigenicity, allergenicity and physiochemical parameter of vaccine construct

The potential of the vaccine construct in eliciting immunological response through binding to the B and T cell receptor was determined by first determining the antigenicity of the vaccine construct. The ANTIGENpro server (scratch.proteomics.ics.uci.edu); a sequence-based alignment free and pathogen independent predictor was used to generate the antigenicity index (*Khatoon, Pandey & Prajapati, 2017*).

The non-allergenic potential of the multi-epitope vaccine construct was determined using the AllerTOP v.2.0 server (www.ddg-pharmfac.net/AllerTOP). Several physiochemical parameters of the vaccine construct which include amino acid composition, the theoretical pI, instability index, in-vitro and in-vivo half-life, aliphatic index, molecular weight and grand average of hydropathicity (GRAVY) parameters were all assessed using the ProtParam server (https://web.expasy.org/protparam/) (*Onile, 2014*).

## Prediction, refinement and validation of the tertiary structure of vaccine construct

The 3D structure of the vaccine construct was predicted using RaptorX structure prediction server (raptorx.uchicago.edu/). This server predicts secondary and tertiary structures as well as contact map, solvent accessibility, disordered regions and binding sites of protein sequences. RaptorX server also assign some confidence scores *P*-value, GDT (global distance test) and uGDT (un-normalized GDT), and modeling error at each

residue to indicate the quality of a predicted 3D mode. In order to improve and refine the predicted 3D model of vaccine construct, the output model of RaptorX server was further modelled using GalaxyRefine server (http://galaxy.seoklab.org/) (*Heo, Park & Seok, 2013*). This server uses CASP10 based refinement approach to reconstruct and repack the protein side chain, followed by dynamics simulation to relax the structure. ProSA-wed (https://prosa.services.came.sbg.ac.at/prosa.php) was employed in the tertiary structure validation; also the refinement output was validated using Ramachandran plot analysis (Mordred.bioc.cam.au.ck/~rapper/rampage.php).

## Disulfide engineering for vaccine stability

The Disulfide by Design v2.12 server was used for the in silico disulfide engineering of the refined model 3D structure of the vaccine construct. The refined model was first run to determine residue pairs useful for the purpose of disulphide engineering. Only residue pairs with energy (kcal/mol) less than 2.2 and Chi$^3$ ($\chi^3$) between −87 and +97 degree were selected for mutation (*Rana & Akhter, 2016*). A total of 5 residue pairs were eventually selected for mutation by using the create mutate icon of the Disulfide by Design server.

## Codon adaptation and in silico cloning

To determine and optimise the expression rate of the vaccine construct in a proper expression vector, the primary sequence of the vaccine construct was inputted into the Java Codon Adaptation Tool (JCAT) for codon optimization which was performed in the host *Escherichia coli* strain K12. Due to difference in codon usage of *E. coli* and native host *S. haematobium* from where the vaccine construct sequences originated, three different options in the JCAT server were selected which include: avoid rho-independent transcription, prokaryotes ribosome binding site and restriction enzymes cleavage sites. To determine the high-level of protein expression, the output of the JCAT which include codon adaptation Index and percentage GC content were used. *E. coli* pET-28(+) was used as vector to clone the adapted nucleotide sequence (provided by JCAT server) of the final vaccine construct using the restriction cloning module of SnapGene tool (*Ali et al., 2017*).

## Molecular docking of vaccine construct with TLR4

The ClusPro 2.0 server was used for the molecular protein–protein docking to check for the binding affinity between the vaccine construct and TLR-4 receptor (*Kozakov et al., 2017*; *Pandey, Bhatt & Prajapati, 2018*). The refined sub unit multi-epitope vaccine protein and TLR-4 PDB file (4G8A) obtained from the RCSB-Protein Data Bank were used as the ligand and receptor respectively.

## RESULTS

### Retrieved *S. haematobium* HSP sequences for vaccine construct

To design the multi-epitope subunit vaccine for urinary schistosomiasis consisted of seven (7) HSP sequences (HSP90 (KGB37337.1), HSP11 (XP_012799478.1), HSP97

**Table 1 Retrieved *S. haematobium* Heat shock protein accession numbers from the NCBI database and their antigenicity scores.**

| Serial no. | Protein accession no. | Protein name | Antigenicity scores | Selected/non-selected |
|---|---|---|---|---|
| 1 | KGB40455.1 | 60 kDa Heat shock protein | 0.669986 | Non-selected |
| 2 | KGB38609.1 | 75 kDa Heat shock protein | 0.665038 | Non-selected |
| 3 | KGB41273.1 | 10 kDa Heat shock protein | 0.608315 | Non-selected |
| 4 | KGB37622.1 | Heat shock protein 83 | 0.665191 | Non-selected |
| 5 | KGB34481.1 | 97 kDa Heat shock protein | 0.926815 | Selected |
| 6 | XP_012799478.1 | Heat shock protein beta-11 | 0.895751 | Selected |
| 7 | KGB37337.1 | Activator of 90 kDa Heat shock protein ATPase 1 | 0.858776 | Selected |

(KGB34481.1), HSP75 (KGB38609.1), HSP83 (KGB37622.1), HSP10 (KGB41273.1), HSP60 (KGB40455.1)) were retrieved from the NCBI database (https://www.ncbi.nlm.nih.gov) in FASTA format. Only three (HSP11, HSP97, HSP90) proteins had an antigenic probability of ≥0.8 after retrieved protein sequences were analyzed using ANTIGENpro for antigenicity prediction (Table 1), consewuently they were selected for the vaccine construct.

## Predicted CTL, HTL epitope and immunogenicity

By using the NetCTL 1.2 server for prediction of the CTL receptor specific immunogenic epitopes, a total of 23 CTL epitopes (9mer length) were obtained for the inputted three HSP protein sequences and only 12 epitopes with high immunogenicity scores as obtained from IEDB server were chosen as the final CTL epitopes to undergo vaccine designing (Table 2).

The prediction of the HTL epitope was carried out using the IEDB MHC-II epitope prediction module, all three protein sequences were subjected to the module. The mouse alleles used for prediction were H2-1Ad, H2-1Ed and H2-1Ab. However, only 17 HTL epitopes with lowest percentile with ≤1.5 with MHC-II were selected and subjected for the final multi-epitope vaccine construct (Table 3A). Also, all the 17 HTL epitopes have the capacity of inducing IFN-γ, and this was evident from the positive score obtained from the IFN epitope server output (Table 3B).

## Constructed multi-epitope subunit vaccine

The constructed multi-epitope subunit vaccine consisted of 12 CTL epitopes and 17 HTL epitopes with high binding affinity fused together with the help of AAY, GPGPG and EAAAK linkers. The adjuvant and the CTL epitopes were combined by the EAAAK linkers, the intra-CTL and intra-HTL epitopes were joined by intra-CTL AAY and intra-HTL GPGPG linker respectively to form the final vaccine constructs composed of 493 amino acid residues. For maximum ability of the candidate vaccine to elicit adequate immune response, adjuvant and CTL epitopes were joined together using an EAAAK linker (Fig. 1A).

**Table 2 Predicted cytotoxic T-lymphocyte (CTL) epitopes and their immunogenicity scores for *S. haematobium* as obtained from the immune epitope database.**

| S/N | Accession ID | Epitopes | Scores | Length | Immunogenicity scores | Selected/non selected |
|-----|--------------|----------|--------|--------|----------------------|----------------------|
| 1 | KGB34481.1 | LDMTEEWLY | 0.7760 | 9 | 0.35427 | Selected |
| | | QTEEIDGTL | 0.9769 | 9 | 0.33501 | Selected |
| | | SIAAGEPTY | 0.8120 | 9 | 0.19264 | Selected |
| | | SQAQLIEEY | 1.3793 | 9 | 0.15362 | Selected |
| | | NSKNAVEEY | 1.7785 | 9 | 0.14345 | Selected |
| | | FTVIEQCLY | 2.7477 | 9 | 0.0838 | Selected |
| | | QLEDMIVQY | 1.8465 | 9 | −0.02374 | Non-selected |
| | | VTDIVSQQQ | 0.8464 | 9 | −0.13985 | Non-selected |
| | | FTEPRKIKL | 0.7619 | 9 | −0.14494 | Non-selected |
| | | FTTKQLNEF | 1.0480 | 9 | −0.2746 | Non-selected |
| | | SIEVSNMQF | 0.8128 | 9 | −0.30903 | Non-selected |
| 2 | KGB39720.1 | GSDGCFVSF | 2.2740 | 9 | 0.03718 | Selected |
| 3 | XP_012797099.1 | FSGNITGIF | 1.0475 | 9 | 0.27699 | Selected |
| | | KLDGEANVY | 2.2518 | 9 | 0.19429 | Selected |
| | | GSKIENDLY | 2.0995 | 9 | 0.16757 | Selected |
| | | EVISLIDEY | 0.7906 | 9 | 0.06843 | Selected |
| | | TSSTDGDLV | 0.9172 | 9 | 0.0511 | Selected |
| | | WSDKDATGW | 1.0050 | 9 | −0.09881 | Non-selected |
| | | YITLLKEDY | 1.1013 | 9 | −0.1149 | Non-selected |
| | | WTSSTDGDL | 0.9051 | 9 | −0.11993 | Non-selected |
| | | LAQKNVPAY | 0.9310 | 9 | −0.20854 | Non-selected |
| | | FLAALKQTY | 1.7104 | 9 | −0.23681 | Non-selected |
| | | AALKQTYGY | 0.7829 | 9 | −0.28018 | Non-selected |

Note:
Only epitopes with Comb score >0.75 were selected for immunogenicity test and CTL epitopes (12) with high immunogenicity score (+ve) were selected and subjected to the vaccine designing.

## Predicted B cell epitopes for *S. haematobium* proteins

A total of 14 epitopes having 0.99 and above probability score with 20mers length were selected as the linear B cell binding epitopes for the final vaccine construct (Table 4). Discontinuous epitopes of 125 amino acids were predicted from the final 3D model of vaccine construct with the score of 0.734 (Fig. 1B).

## Predicted antigenicity, allergenicity and physiochemical parameter of the vaccine construct

The antigenicity test of the designed vaccine construct showed the antigenicity probability to be 0.905150, this represents the antigenic nature of the vaccine construct. An allergenicity test carried out for the vaccine construct also showed that the vaccine candidate is non-allergenic in nature.

The prediction of the physiochemical parameters of the vaccine construct was characterized by using the ProtParam server which evaluated for the following parameters;

**Table 3 (A)** Predicted helper T-lymphocyte (HTL) epitopes and their percentile ranks for *S. haematobium* as obtained from the immune epitope database. **(B)** Interferon gamma (IFN) Inducing capacity test for the predicted HTL epitopes for urinary schistosomiasis.

| (A) Allele | Seq. no. | Start | End | Epitope | Method | Percentile rank* |
|---|---|---|---|---|---|---|
| H2-IAd | 2 | 54 | 68 | RKVSQIKIVTSRVKA | Consensus (smm/nn) | 0.38 |
| H2-IAd | 2 | 53 | 67 | PRKVSQIKIVTSRVK | Consensus (smm/nn) | 0.47 |
| H2-IAd | 2 | 52 | 66 | QPRKVSQIKIVTSRV | Consensus (smm/nn) | 0.48 |
| H2-IAd | 3 | 435 | 449 | RSLVRIRKVLDSIAA | Consensus (smm/nn) | 0.64 |
| H2-IAd | 3 | 500 | 514 | VSQQQAMESVCHPII | Consensus (smm/nn) | 0.91 |
| H2-IAd | 3 | 498 | 512 | DIVSQQQAMESVCHP | Consensus (smm/nn) | 0.94 |
| H2-IAd | 3 | 499 | 513 | IVSQQQAMESVCHPI | Consensus (smm/nn) | 1.03 |
| H2-IAd | 3 | 434 | 448 | NRSLVRIRKVLDSIA | Consensus (smm/nn) | 1.04 |
| H2-IAd | 3 | 100 | 114 | CAFQAAICSPAFKVK | Consensus (smm/nn) | 1.05 |
| H2-IAd | 2 | 51 | 65 | DQPRKVSQIKIVTSR | Consensus (smm/nn) | 1.12 |
| H2-IAd | 3 | 433 | 447 | FNRSLVRIRKVLDSI | Consensus (smm/nn) | 1.21 |
| H2-IAd | 3 | 99 | 113 | GCAFQAAICSPAFKV | Consensus (smm/nn) | 1.23 |
| H2-IAd | 3 | 432 | 446 | NFNRSLVRIRKVLDS | Consensus (smm/nn) | 1.27 |
| H2-IAd | 3 | 98 | 112 | RGCAFQAAICSPAFK | Consensus (smm/nn) | 1.32 |
| H2-IAd | 1 | 214 | 228 | YRVLTTKELVKAFTR | Consensus (smm/nn) | 1.35 |
| H2-IAd | 1 | 289 | 303 | TRLFLAQKNVPAYDL | Consensus (smm/nn) | 1.37 |
| H2-IAd | 3 | 497 | 511 | TDIVSQQQAMESVCH | Consensus (smm/nn) | 1.4 |

| (B) Serial no. | Epitope | Method | Result | Score |
|---|---|---|---|---|
| Epitope_1 | RKVSQIKIVTSRVKA | MERCI | POSITIVE | 1 |
| Epitope_2 | PRKVSQIKIVTSRVK | MERCI | POSITIVE | 1 |
| Epitope_3 | QPRKVSQIKIVTSRV | SVM | POSITIVE | 0.57919869 |
| Epitope_4 | RSLVRIRKVLDSIAA | MERCI | POSITIVE | 6 |
| Epitope_5 | VSQQQAMESVCHPII | MERCI | POSITIVE | 1 |
| Epitope_6 | DIVSQQQAMESVCHP | MERCI | POSITIVE | 1 |
| Epitope_7 | IVSQQQAMESVCHPI | MERCI | POSITIVE | 1 |
| Epitope_8 | NRSLVRIRKVLDSIA | MERCI | POSITIVE | 8 |
| Epitope_9 | CAFQAAICSPAFKVK | MERCI | POSITIVE | 1 |
| Epitope_10 | DQPRKVSQIKIVTSR | SVM | POSITIVE | 0.56096058 |
| Epitope_11 | FNRSLVRIRKVLDSI | MERCI | POSITIVE | 9 |
| Epitope_12 | GCAFQAAICSPAFKV | MERCI | POSITIVE | 1 |
| Epitope_13 | NFNRSLVRIRKVLDS | MERCI | POSITIVE | 8 |
| Epitope_14 | RGCAFQAAICSPAFK | MERCI | POSITIVE | 2 |
| Epitope_15 | YRVLTTKELVKAFTR | MERCI | POSITIVE | 11 |
| Epitope_16 | TRLFLAQKNVPAYDL | MERCI | POSITIVE | 1 |
| Epitope_17 | TDIVSQQQAMESVCH | SVM | POSITIVE | 0.50508595 |

**Note:**
* Only HTL epitopes with higher binding affinity (≤1.5) with MHC-II were selected for the final multi-epitope vaccine construct.

The molecular weight was 51409.68 kDa, theoretical pI 9.11, while the total number of negative and positive charge residues were 36 and 49 respectively. The estimated half-life was 4.4 h in mammalian reticulocytes in-vitro, 20 and 10 h in yeast and *E. coli* respectively.

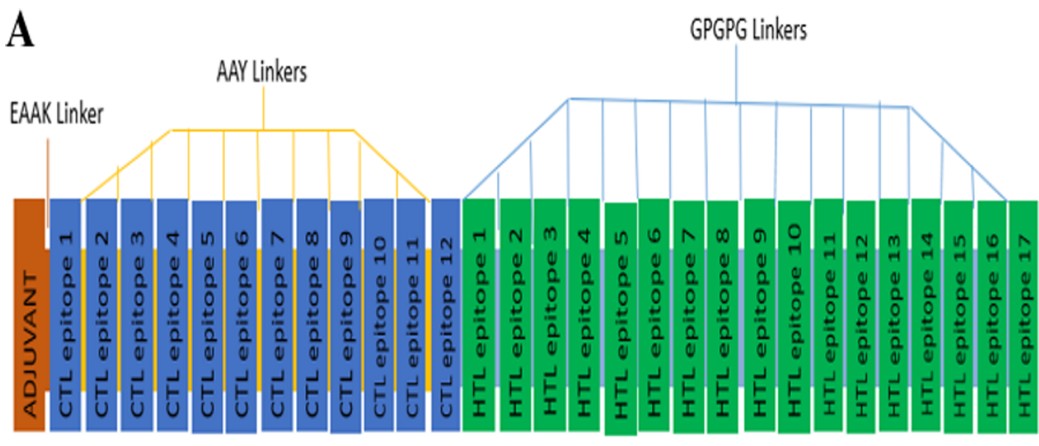

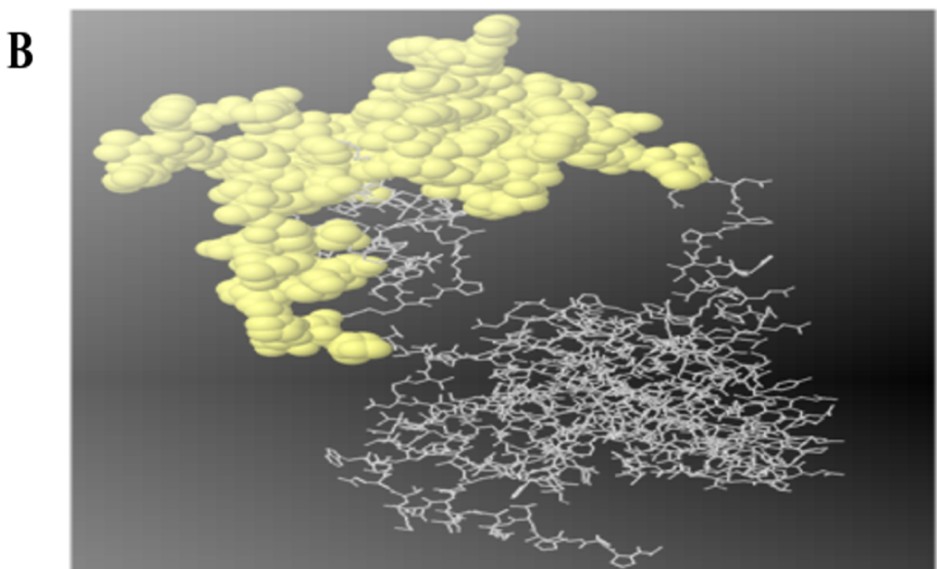

**Figure 1** (A) Diagram of final vaccine construct. The multi-epitope vaccine sequence consisting of 493 amino acid residues; of which the adjuvant and CTL epitope has been joined by EAAK linker, where AAY and GPGPG linkers were used to join the CTL and HTL epitopes, respectively. (B) Conformation of B-cell epitopes (yellow colour) showing the sequence subunits composed of antigenic epitopes that will come in direct contact with immune receptor.

The extinction coefficient was found to be 36790 $M^{-1}cm^{-1}$, at 280 nm measured in water assuming all pairs of cysteine residues are reduced, instability index was given to be 35.01. The value of the aliphatic index was 77.81 while the grand average of hydropathicity (GRAVY) was −0.138.

## Predicted, refined and validated tertiary structure of vaccine construct

The 3D model and tertiary structure were predicted using RaptorX server (Fig. 2). The best template for the homology modeling was PDB ID: 5ja1A with 493 (100%) amino acid residues modelled as single domain with 4% disorder. The secondary structure information includes helix 31%, Beta sheet 17% and coiled structure 51%. To refine the vaccine construct, GalaxyRefine server was used. Out of all refined models, model 3 was

**Table 4 B cell specific epitopes and their score as predicted and obtained from the immune epitope database.**

| S/N | Position | Epitope | Score |
|---|---|---|---|
| 1 | 170 | RVKAGPGPGPRKVSQIKIVT | 1 |
| 2 | 247 | SVCHPIIGPGPGDIVSQQQA | 1 |
| 3 | 370 | LDSIGPGPGGCAFQAAICSP | 1 |
| 4 | 328 | PAFKVKGPGPGDQPRKVSQI | 1 |
| 5 | 285 | AMESVCHPIGPGPGNRSLVR | 1 |
| 6 | 411 | LDSGPGPGRGCAFQAAICSP | 1 |
| 7 | 191 | RVKGPGPGQPRKVSQIKIVT | 1 |
| 8 | 145 | TSSTDGDLVGPGPGRKVSQI | 1 |
| 9 | 307 | KVLDSIAGPGPGCAFQAAIC | 1 |
| 10 | 226 | KVLDSIAAGPGPGVSQQQAM | 1 |
| 11 | 442 | LTTKELVKAFTRGPGPGTRL | 1 |
| 12 | 349 | IVTSRGPGPGFNRSLVRIRK | 1 |
| 13 | 467 | NVPAYDLGPGPGTDIVSQQQ | 1 |
| 14 | 54 | IEEYAAYNSKNAVEEYAAYF | 0.994 |
| 15 | 14 | DMTEEWLYAAYQTEEIDGTL | 0.753 |

**Note:**
Only 20mers epitopes with equal and above 0.99 score were selected.

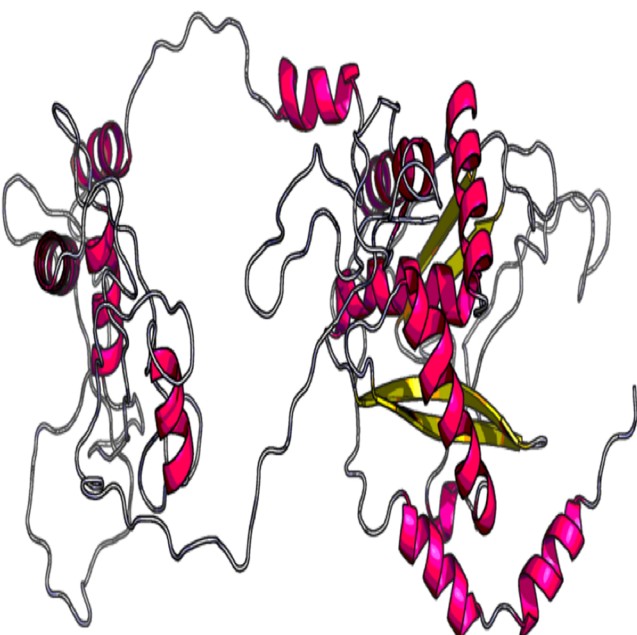

**Figure 2 Tertiary structure of predicted vaccine construct showing the helix, sheet and coiled region.**

selected as the best based on various parameters including GDT-HA (0.9239), RSMD (0.484), MolProbity (2.69), Clash Score (32.7) and Poor rotamers (1.9).

The refinement output was also validated using Ramachandran plot analysis, which revealed that 93.3% of the residue are located in most favoured regions, 5.3% in allowed

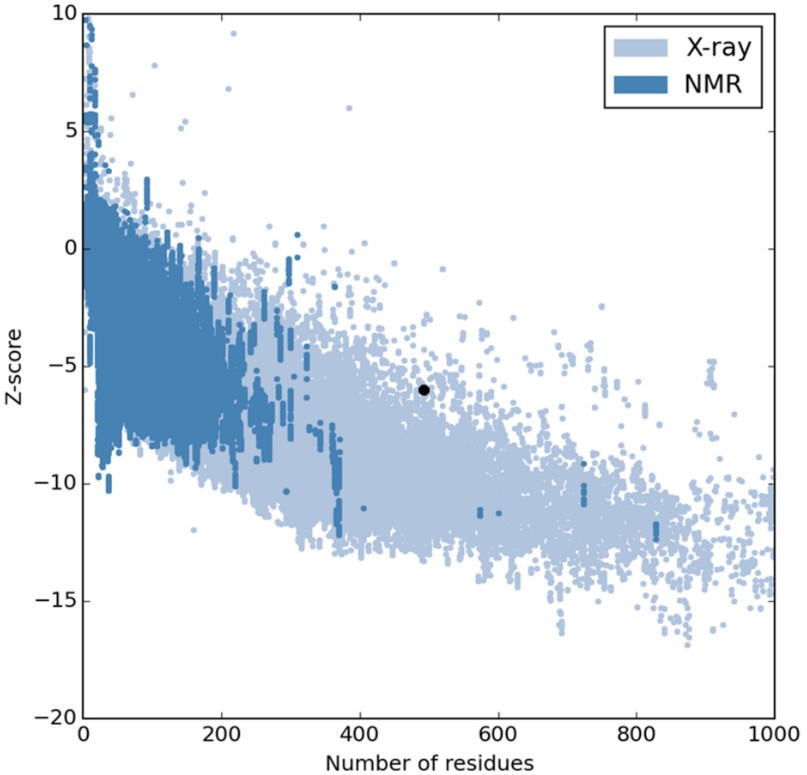

**Figure 3 PROSA validation of 3D vaccine structure showing Z-score (−6.01).**

regions and only 1.4% in outlier region and the quality and potential errors in the crude 3D as verified by ProsSA-web showed Z-scores of −6.01 for ProSA (Figs. 3 and 4).

## Disulfide engineering for vaccine stability

Disulfide engineering was done to stabilize the final model of the vaccine construct using Disulfide by Design v2.12. The result showed a total 58 pairs of residues that are useful for the purpose of disulphide engineering (Fig. 5) but 10 mutations were created from the only five pairs of residues (GLU56-CYS249, TYR60-HIS493, VAL116-GLN242, GLN243-GLU489) that were found with energy value less than 2.2 and Chi3 value range between −87 and +97 degree.

## Codon adaptation and in silico cloning

The codon representing the subunit vaccine candidate was adapted to the codons of the *E. coli* K 12 strain using JCAT server and showed that the optimized codon sequence has a length of 1,450 nucleotides with a codon adaptation index (CAI) of 0.95 and average GC content of 54.56%. The adapted codon sequence was later inserted into pET28(+) vector to form the restriction clone (Fig. 6). The adapted codon sequences used in insilico cloning are made up of the following 1,400 nitrogenous bases "GCTCCGCCGCACGC TCTGTCTGAAGCTGCTGCTAAACTGGACATGACCGAAGAATGGCTGTACGCTG CTTACCAGACCGAAGAAATCGACGGTACCCTGGCTGCTTACTCTATCGCTGC
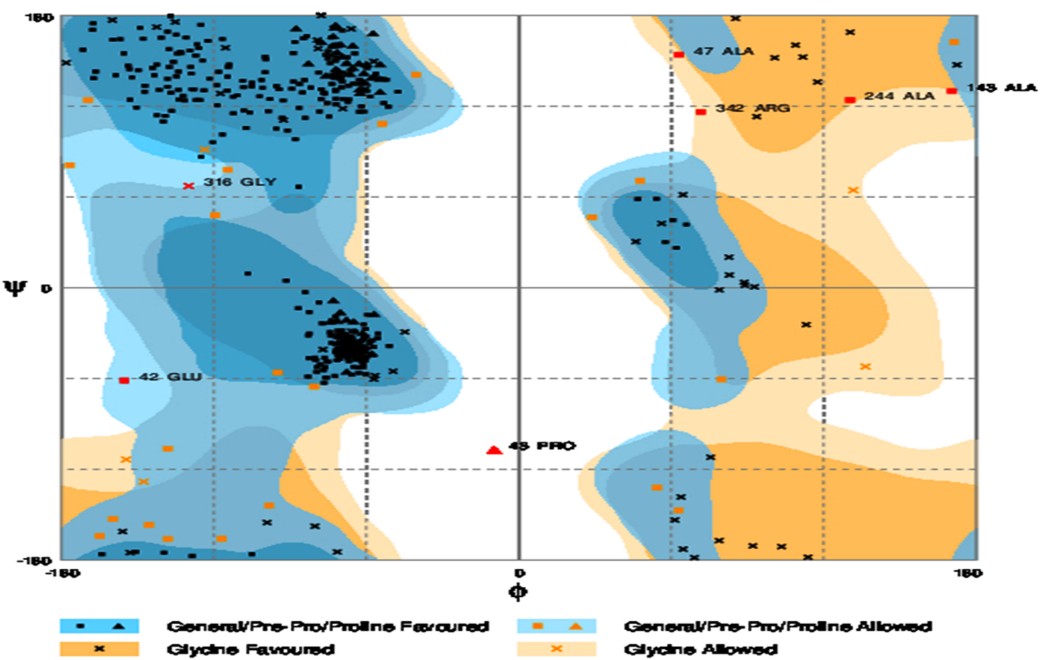

**Figure 4 Ramachandran plot formation to the validated 3D modelled structure of vaccine protein.**
The plot showed 93.3% residues of vaccine protein in favoured region.

TGGTGAACCGACCTACGCTGCTTACTCTCAGGCTCAGCTGATCGAAGAATACG
CTGCTTACAACTCTAAAAACGCTGTTGAAGAATACGCTGCTTACTTCACCGTT
ATCGAACAGTGCCTGTACGCTGCTTACGGTTCTGACGGTTGCTTCGTTTCTT
TCGCTGCTTACTTCTCTGGTAACATCACCGGTATCTTCGCTGCTTACAAACT
GGACGGTGAAGCTAACGTTTACGCTGCTTACGGTTCTAAAATCGAAAACGA
CCTGTACGCTGCTTACGAAGTTATCTCTCTGATCGACGAATACGCTGCTTAC
ACCTCTTCTACCGACGGTGACCTGGTTGGTCCGGGTCCGGGTCGTAAAGTTTC
TCAGATCAAAATCGTTACCTCTCGTGTTAAAGCTGGTCCGGGTCCGGGTCCG
CGTAAAGTTTCTCAGATCAAAATCGTTACCTCTCGTGTTAAAGGTCCGGGTCC
GGGTCAGCCGCGTAAAGTTTCTCAGATCAAAATCGTTACCTCTCGTGTTGG
TCCGGGTCCGGGTCGTTCTCTGGTTCGTATCCGTAAAGTTCTGGACTCTATC
GCTGCTGGTCCGGGTCCGGGTGTTTCTCAGCAGCAGGCTATGGAATCTGTT
TGCCACCCGATCATCGGTCCGGGTCCGGGTGACATCGTTTCTCAGCAGCAG
GCTATGGAATCTGTTTGCCACCCGGGTCCGGGTCCGGGTATCGTTTCTCAG
CAGCAGGCTATGGAATCTGTTTGCCACCCGATCGGTCCGGGTCCGGGTAACCG
TTCTCTGGTTCGTATCCGTAAAGTTCTGGACTCTATCGCTGGTCCGGGTCCG
GGTTGCGCTTTCCAGGCTGCTATCTGCTCTCCGGCTTTCAAAGTTAAAGGTCC
GGGTCCGGGTGACCAGCCGCGTAAAGTTTCTCAGATCAAAATCGTTACCTCTC
GTGGTCCGGGTCCGGGTTTCAACCGTTCTCTGGTTCGTATCCGTAAAGTTCTG
GACTCTATCGGTCCGGGTCCGGGTGGTTGCGCTTTCCAGGCTGCTATCTGCTC
TCCGGCTTTCAAAGTTGGTCCGGGTCCGGGTAACTTCAACCGTTCTCTGGTT
CGTATCCGTAAAGTTCTGGACTCTGGTCCGGGTCCGGGTCGTGGTTGCGCTTT

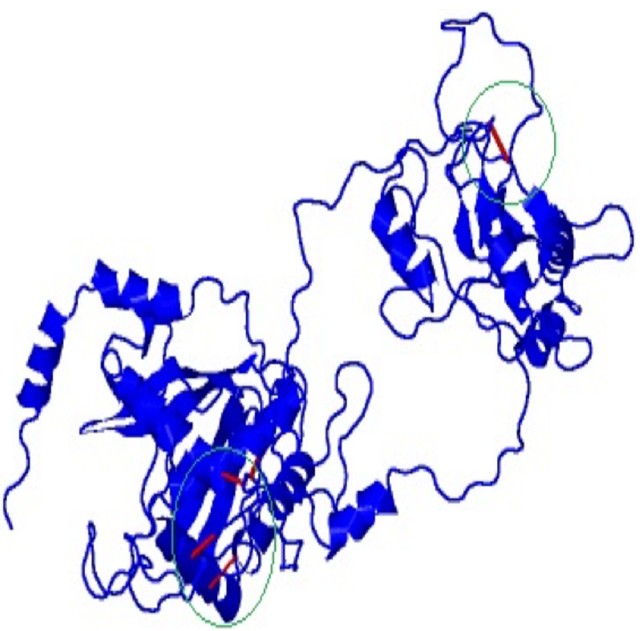

**Figure 5 Mutated and stabilized vaccine construct protein by disulphide engineering.** The engineered protein showed the disulfide bonds in red colour as presented by the mutated platform of Disulfide by Design server.

CCAGGCTGCTATCTGCTCTCCGGCTTTCAAAGGTCCGGGTCCGGGTTACCGT GTTCTGACCACCAAAGAACTGGTTAAAGCTTTCACCCGTGGTCCGGGTCCGGG TACCCGTCTGTTCCTGGCTCAGAAAAA".

## Molecular docking of vaccine construct with TLR4

A total of 29 models were generated from the molecular protein–protein docking between the vaccine construct and the TLR4 receptor using ClusPro 2.0 server. Only one model (000.23) with the lowest energy score (−1,250.0 and highest binding affinity among other predicted docked complex fulfilled the desired criteria for best-docked complex and was eventually selected (Fig. 7).

## DISCUSSION

Praziquantel (PZQ) is a pyrazinoisoquinole antihelminthic drug which is one of the few commercially available anti-schistosoma drugs with few known side effects (*Aragon et al., 2009*). The report of possible drug resistance to praziquantel due to drug pressure has now become of huge public health concern. Several studies have reported remarkable cases of resistance to PZQ which include the development of heritable trait that maintained drug resistance to PZQ in more than six generations (*Devaney, 2006*; *Doenhoff, Cioli & Utzinger, 2008*; *Fallon et al., 1996*; *Masamba et al., 2016*; *William et al., 2001*). Only 13% of targeted population received praziquantel treatment, with the drug not capable of preventing reinfection, requires repeated treatment and its characterised by reduction in efficiency among population with heavy infection (*Tallima et al., 2017*). With the endemicity of schistosomiasis still on the increase across many countries, with increased

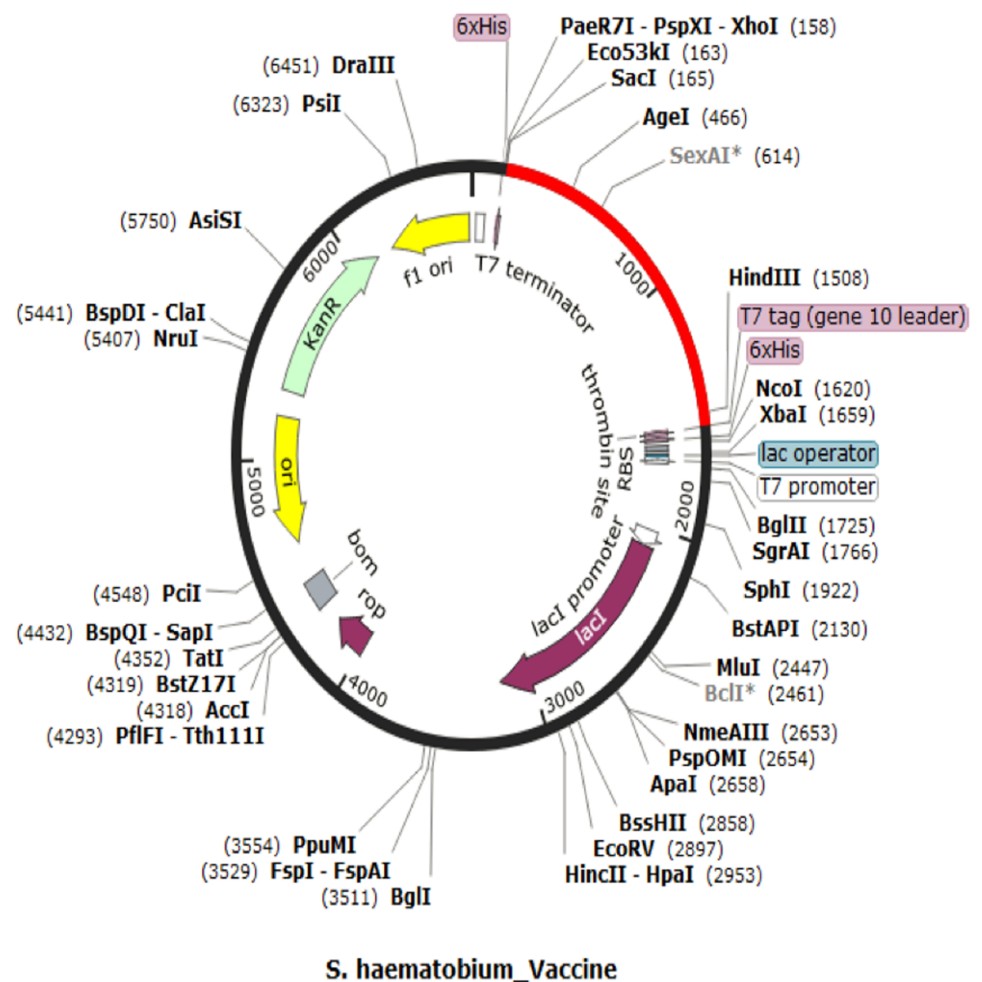

**Figure 6 In silico restriction cloning for adapted vaccine construct into pET28a(+) expression vector.** The adapted vaccine sequences is shown in red colour as the region of insert while the black circle represent the vector.

cases of Schistosoma resistance to the most used and reliable drug PZQ. It is now imperative to change the direction of treatment and focus on a lasting line of treatment by targeting the parasitic lifecycle and its different developmental stages, which could well provide a lasting solution to the spread of infection (*Masamba et al., 2016*). The current study focused on retrieving seven (7) *S. haematobium* HSP sequences for the design of possible multi-epitope subunit vaccine (logistically feasible and safely profiled as described by *Shey et al. (2019)*) construct for the treatment of urinary schistosomiasis using computational approaches. Epitopes-based vaccine will avoid responses against other unsuitable epitopes on antigen, thereby generating more specific immune response against the antigen under consideration (*Shey et al., 2019*). Only three of these seven HSPs were eventually selected for the design of vaccine due to their predicted high antigenic potential (*Khatoon, Pandey & Prajapati, 2017*). *Masamba et al. (2016)* has proposed that selective identification of small molecule inhibitors or peptides as an inhibitor of HSPs

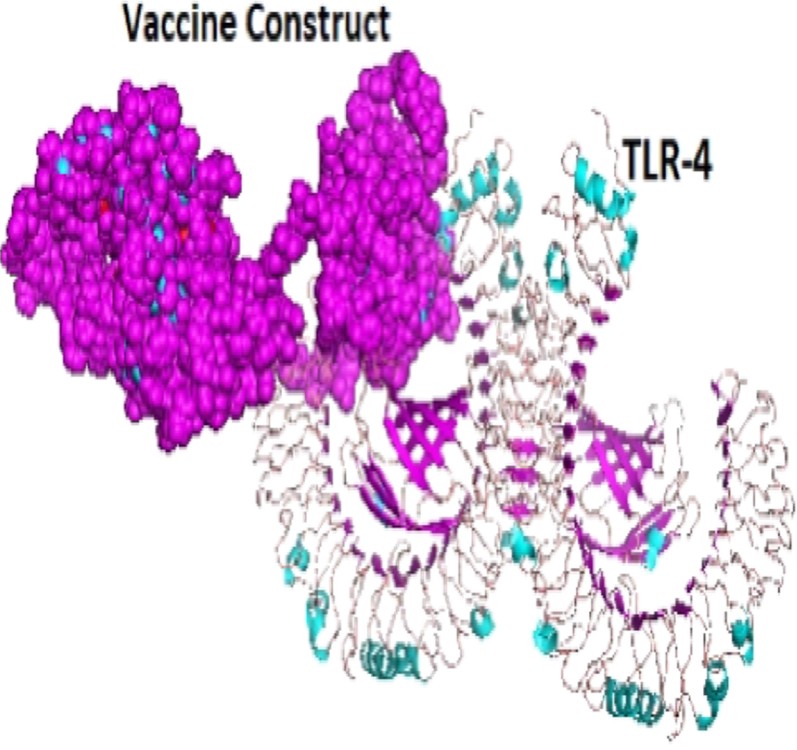

**Figure 7 The docked complex showing the vaccine protein (dot) and TLR-4 receptor (cartoon) interaction.**

at the schistosomula stage could be useful in targeting and preventing re-infection of schistosomiasis. Parasites use several mechanisms to upregulate Universal Stress Proteins (including HSPs) which assist the parasite to tolerate different environmental and adverse conditions especially at the schistosomula stage of parasite development (*Mbah et al., 2013*). The HSPs CTL and HTL epitopes with predicted high immunogenicity were selected for inclusion in the vaccine construct. Thus the final vaccine construct had more of high affinities for MHC Class I, II and B-cell epitopes. The CTL-based response targets cells having intracellular viral, bacterial or protozoan infection (*Jordan & Hunter, 2010*) while the HTL-based response is essential for both humoral and cell-mediated immune response (*Pross & Lefkowitz, 2008*). Activated CD8+ cytotoxic T cells are reportedly able to promote death of parasite carrying host MHC class I molecules in its surface (*Zhou et al., 2012*). Also mice immunized with *S. japonicum* 22.6/26GST coupled to Sepharose 4B bead was able to reduce the parasite burden due to an increase in the number of activated CD8+ cells (*Zhou et al., 2012*). Several studies have used mouse model to understand host protective immune response against schistosome infections (*S. mansoni*) and both antibodies and T cells are needed to ensure maximum protection (*Colley & Secor, 2014*; *Jankovic et al., 1999*). Two previously reported spacer sequences (AAY and GPGPG linkers) were used to improve the designed vaccine (*Shey et al., 2019*) by incorporating between the predicted CTL and HTL epitopes respectively, thereby

producing sequences with minimal junctional immunogenicity leading to the design of a more potent multi-epitope vaccine (*Meza et al., 2017*; *Shey et al., 2019*).

The allergenicity and antigenicity test of the designed vaccine predicted its safety for human use and also revealed its potency to inhibit the entrance of Schistosomes into the human host (*Ali et al., 2017*; *Khatoon, Pandey & Prajapati, 2017*; *Pandey, Bhatt & Prajapati, 2018*). Using physiochemical parameters and structural features, the designed vaccine weighed is 51 kDa, which is an average molecular weight for a multi-subunit vaccine (*Khatoon, Pandey & Prajapati, 2017*). This therefore favours the antigenicity of the vaccine (*Khatoon, Pandey & Prajapati, 2017*; *Pandey, Bhatt & Prajapati, 2018*). The theoretical pI of 9.11 showed that the designed vaccine is basic in nature; the obtained higher aliphatic index score of 77.81 classified the vaccine protein as thermostable (*Ali et al., 2017*; *Khatoon, Pandey & Prajapati, 2017*), while the negative value of the Grand average of hydropathicity of the vaccine represents its hydrophilic nature.

The 3-dimensional structure obtained from the highly accurate homology modeling servers (RaptorX and GalaxyRefine) contain sufficient information about the arrangement of protein amino acid residues, percentage disorder and secondary structures. Also, the Ramachandran plot showed very few outliner residues with most of the residues clustering at very favourable regions thereby ratifying the quality of the overall model as satisfactory. This information was useful in the study of vaccine construct function, dynamics and interaction with ligands and other proteins (*Khatoon, Pandey & Prajapati, 2017*).

To enhance the thermostability of the designed vaccine, novel disulphide bonds were further introduced into the multi-epitope subunit vaccine protein (*Rana & Akhter, 2016*). In order to express the vaccine construct in the *E. coli* expression system, an in silico cloning was done (*Khatoon, Pandey & Prajapati, 2017*; *Pandey, Bhatt & Prajapati, 2018*). Codon adaptation of the designed vaccine was done according to the usage of *E. coli* (strain K12) expression system using JCAT server and found the codon adaptive index to be 0.95 whereas, a score close to 1.0 was described elsewhere as satisfactory (*Pandey, Bhatt & Prajapati, 2018*). A protein–protein docking analysis was carried out to determine the immune response of TLR-4 agonists against the multi-epitope subunit vaccine designed by minimizing the potential energy of the complete system. The energy minimization ensured conformational stability of designed vaccine-TLR-4 complex by repairing the unnecessary geometry of the structure (*Pandey et al., 2017*; *Pandey et al., 2015*). *El-Din (2016)* has reported decrease in *S. mansoni worm* load, egg load and granuloma size during TLR4 and TLR 9 stimulation with upregulated expression of macrophages. TLR 4 is reportedly known to play significant role in the dentritic cells and macropahges recognition of helminthes products; macrophages production of cytokines and development of Th2 responses (*El-Din, 2016*; *Kane, Jung & Pearce, 2008*). In a Th1/Th2 cytokine secretion assay and DNA microarray anaylysis, activated T cells, up regulation of some cytotoxic genes, followed by increase in parasite egg load have been reported in mice deficient in TLR (*Zhang et al., 2011*).

## CONCLUSION

In this study, we used several immunoinformatics approaches to design vaccine thay may be effective against urinary schistosomiasis. Sequences of *S. haematobium* HSPs were

retrieved from the database to design a multi-epitope subunit vaccine containing a CTL, HTL and BCL epitopes of varying length. The designed vaccine was stable with high antigenic properties, high binding affinity for TLR-4 receptor and was found to be non-allergic for human use. However, the designed vaccine requires experimental validation in order to establish it ability in controlling Schistosoma infection through the generation of effective immune response and memory.

### Funding

The authors received no funding for this work.

### Competing Interests

The authors declare that they have no competing interests.

### Author Contributions

- Olugbenga S. Onile conceived and designed the experiments, performed the experiments, analyzed the data, prepared figures and/or tables, authored or reviewed drafts of the paper, and approved the final draft.
- Adeyinka I. Fadahunsi performed the experiments, analyzed the data, prepared figures and/or tables, authored or reviewed drafts of the paper, and approved the final draft.
- Ameerah A. Adekunle conceived and designed the experiments, performed the experiments, analyzed the data, prepared figures and/or tables, authored or reviewed drafts of the paper, and approved the final draft.
- Bolaji F. Oyeyemi conceived and designed the experiments, analyzed the data, authored or reviewed drafts of the paper, and approved the final draft.
- Chiaka I. Anumudu analyzed the data, authored or reviewed drafts of the paper, and approved the final draft.

### Data Availability

The HSP sequences available at NCBI: HSP90 (KGB37337.1), HSP11 (XP_012799478.1), HSP97 (KGB34481.1), 212 HSP75 (KGB38609.1), HSP83 (KGB37622.1), HSP10 (KGB41273.1), HSP60 (KGB40455.1).

Additional data used in this study is available at the RCSB-Protein Data Bank: search term 4G8A.

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
