# Peer review of "An immunoinformatics approach for the design of a multi-epitope subunit vaccine for urogenital schistosomiasis"

_PeerJ, doi:10.7717/peerj.8795_

## Round 0.1 · original submission · Major Revisions

Please use all the reviewers' comments to improve the manuscript. In particular, it is essential that the limitations of the study are clearly stated.

Reviewer 1 ·

Basic reporting

This is an interesting study aiming to provide insights on a potential effective vaccine against urinary schistosomiasis through an immune-informatic approach. In order to be suitable for publication, several points should be addressed and improved.
Some English editing is required. Some specific points should be better explained and emphasized (see general comments).

Experimental design

In terms of specific methodology and results, the paper is quite clear. But the authors should add some more general information and overview to make the importance of their findings more understandable for a more general readership, in my opinion.

Validity of the findings

The findings are clear. The authors should discuss more the limitations of their study approach.

Additional comments

This is an interesting study aiming to provide insights on a potential effective vaccine against urinary schistosomiasis through an immune-informatic approach. In order to be suitable for publication, several points should be addressed and improved.

In the introduction, the authors mention that “the spread of the disease to regions that were non-endemic before”. I recommend the authors to little emphasize and expand this point here. Indeed, the reader could realize that urinary schistosomiasis is currently becoming a medical issue that is not limited to Africa, but also Europe is facing this disease for several reasons (refer to: J Immigr Minor Health. 2016 Oct;18(5):1237-1240. doi: 10.1007/s10903-015-0272-3; & Infection. 2019 Jun;47(3):395-398. doi: 10.1007/s15010-018-1244-z.). This point may provide additional interest in this paper for a more general readership.

The methodology is quite clear and well-explained. Again, a short introduction/paragraph (before the first subsection) explaining/summarizing the general study approach and aims, might increase the general readership and understanding, considering the scope of this specific journal.

The results are well-organized. However, in order to appreciate the relevance of these finding, it may be useful to provide at some point (in the introduction or discussion) some general aspects about the protective adaptive immune response against S. hematobium in humans. Indeed, in the conclusion, the authors mention “the generation of effective immune response and memory”.

It is not clear the importance of TLR4 in this specific infection and, as a consequence, the relevance of the finding related to “The energy minimization ensured conformational stability of vaccine construct-TLR-4 complex”. I think this discussion should be expanded and, in general, the authors should provide comments and explanation that can make this article be accessible to all journal readership.

The references must be updated and improved according to the general suggestions above, in order to make this study more accessible and interesting. Definitely, both the introduction and discussion should be expanded.

The authors declare they received no funding for this research. However, this seems to be a time-consuming study and require some specific software. Was that part of a more general project? Can they provide any clarifications about this point?

Reviewer 2 ·

Basic reporting

The manuscript title “An immunoinformatics approach for the design of a multiepitope subunit vaccine for urogenital schistosomiasis” by Olugbenga et al, presented the computational serial analysis of pathogenic to predicted/prioritize the potential candidates multiepitope subunit (targeting Heat Shock Proteins (HSPs) vaccine design, an aim to get effective immune response against urogenital schistosomiasis infections. The subunit vaccine construct/design (493 aa) is then checked in silico for antigenicity, immunogenicity, allergenicity and physicochemical parameters analysis, followed by structural analysis (3D) for stability and allergenicity purposes. The disease is important when it comes to In Sub-Saharan Africa with significant mortality rate, hence the subject is of interest to many.

Experimental design

The study design is straightforward and presented in a systematic manner. however, the methodology may be improved.

Validity of the findings

The prediction are purely computational and required experimental validations.

Additional comments

1. The authors focused only on Heat shock proteins (HSP11, HSP97, HSP90) only, what are other good candidates to be targeted.
2. HSP from S. haematobium is retrieved only, while there are three main species infecting humans including S. haematobium, S. japonicum, and S. mansoni. Does the HPS selected are homologous and the sequences of the epitopes are conserved among the species.
3. Does the epitopes are ordered/aligned randomly without prior analysis of their structure confirmation and compatibility of linking, otherwise, it would be having been better to have a proper rationale behind the order of the epitopes. Same is already been done in various publications.
4. The term vaccine construct or construction may be too ambitious to claim at this point, it may be replacing with vaccine design.
5. The size of the poly-epitopic structure is still big as it consists of too 29 epitopes, it may of concern during In-vivo trials, is it possible to design multiple ((2 or more) structures/design from the selected epitopes, or if there is possibility of reducing the number of epitopes in the design and only those selected which shows the maximum antigenicity and immunogenicity.

Reviewer 3 ·

Basic reporting

1. The submitted manuscript requires extensive editing. For example, in the Abstract, Line 30, “eliciting long time immunity” should be changed to “eliciting long-term immunity.” On Line 58, “200 million people are estimated to affect by urogenital” should be changed to “200 million people are estimate to be affected by urogenital.” On Line 67 – 68, “drug resistance to praziquantel due to continuous pressure on drug have” should be changed to something like “drug resistance to praziquantel due to pressure from continuous mass drug administration.” The sentence on Line 84 is a bit confusing. Changing it to: “HSPs were chosen in this study as the target for the development of a new multi-epitope subunit vaccine against urinary schistosomiasis for their roles in schistosome biology” would make it clearer. Line 166 has an unnecessary period after “vector.” In Line 169, “emanated” should be changed to “originated.” There is an incomplete sentence on Lines 264 – 266. This list is not exhaustive and other errors can be found throughout the submitted manuscript.
2. The introduction and background can be improved. For example, the study referencing Bergquist is incorrectly labelled (5 in the introduction and 6 in the References section). Reference 9 cited on Line 65 refers to a study that focuses on oxamniquine with brief mentions to praziquantel. I would recommend that another paper or review be chosen. Reference 18 needs to be changed, as it does not mention the change of environments for the schistosomes. Reference 20 does not mention vaccines nor anti-cancer drugs; they suggest that new anti-schistosome drug targets could be found through targeting Hps70 signaling. Another reference would be more appropriate.
3. The overall structure of the manuscript is standard and no changes are recommended. Figure 1b does not show the entire protein and conformational epitope. Increasing the resolution on Figures 3b and 4 is recommended. Data and analyses have been shared and methods are easy to follow.
4. Overall, the manuscript is self-contained with relevant results to address the gaps in knowledge expressed within the manuscript.

Experimental design

This research article fits the scope of PeerJ. The objective of the authors is defined, relevant, and meaningful. The methods described are easy to follow with sufficient detail and information. However, I would like to see more explanation and rationale behind the adjuvant, linker, and expression vector choices. These selections seem too similar to References 23 and 24. I think that following their example can be useful to start but I would like to hear your own reasoning behind these choices.

Validity of the findings

Your conclusions states that you have "designed an effective vaccine against urinary schistosomiasis using several immunoinformatics approaches." I would reword this to state that you have used immunoinformatics approaches to design a vaccine candidate that *may* be effective.

Additional comments

The submitted manuscript represents an important step in vaccine development. In silico tools can greatly aid in predicting the efficacy of putative epitope-based vaccines. While this vaccine candidate remains to be tested, the authors have outlined a rational approach to its design.

---

## Round 0.2 · Minor Revisions

The reviewers' comments have been addressed but the PeerJ Section Editor had some additional comments that need to be attended to before the manuscript can be accepted:

'While I agree with the acceptance of the manuscript, I think some minor revisions are in order because the relevance of the in silico cloning used is not readily apparent. Did the authors prepare the construct in the wet lab or "simply" optimized the gene sequence towards better predicted performance? If the latter, I think they should provide the full sequence of the optimized clone, otherwise that portion of the manuscript would not be helpful for anyone else.'

Please address these questions in your next revision.

Thanks.

Reviewer 1 ·

Basic reporting

Figures and manuscript are appropriate.

Experimental design

Clear enough.

Validity of the findings

No major concerns.

Additional comments

The authors appropriately addressed my previous comments. No further observations or concerns.

Reviewer 2 ·

Basic reporting

The authors have tried to address the comments.

Experimental design

The authors have tried to address the comments.

Validity of the findings

The authors have tried to address the comments.

Additional comments

The authors have tried to address the comments.

---

## Round 0.3 · accepted · Accept

The manuscript is accepted.